# Multi-Scale Discrete Cosine Transform Network for Building Change Detection in Very-High-Resolution Remote Sensing Images

**Yangpeng Zhu \*, Lijuan Fan, Qianyu Li and Jing Chang**

School of Economics and Management, Xi'an Shiyou University, Xi'an 710065, China;
22211091243@stumail.xsyu.edu.cn (L.F.); 21211091049@xsyu.edu.cn (Q.L.); changj@xsyu.edu.cn (J.C.)
\* Correspondence: zyp@xsyu.edu.cn

**Abstract:** With the rapid development and promotion of deep learning technology in the field of remote sensing, building change detection (BCD) has made great progress. Some recent approaches have improved detailed information about buildings by introducing high-frequency information. However, there are currently few methods considering the effect of other frequencies in the frequency domain for enhancing feature representation. To overcome this problem, we propose a multi-scale discrete cosine transform (DCT) network (MDNet) with U-shaped architecture, which is composed of two novel DCT-based modules, i.e., the dual-dimension DCT attention module ($D^3AM$) and multi-scale DCT pyramid (MDP). The $D^3AM$ aims to employ the DCT to obtain frequency information from both spatial and channel dimensions for refining building feature representation. Furthermore, the proposed MDP can excavate multi-scale frequency information and construct a feature pyramid through multi-scale DCT, which can elevate multi-scale feature extraction of ground targets with various scales. The proposed MDNet was evaluated with three widely used BCD datasets (WHU-CD, LEVIR-CD, and Google), demonstrating that our approach can achieve more convincing results compared to other comparative methods. Moreover, extensive ablation experiments also present the effectiveness of our proposed $D^3AM$ and MDP.

**Keywords:** building change detection; frequency; discrete cosine transform; attention; remote sensing images

## 1. Introduction

With the continuous improvement in the spatial resolution of remote sensing images (aerial images and satellite images), various application requirements for Earth observation are constantly being raised, especially for the observation of finer ground objects, such as buildings [1,2]. This is because buildings are among the essential products of human beings' continuous transformation of the Earth's surface for production and life [3]. The changes in buildings in areas of interest are studied over time. Change detection (CD) is a technique that has been widely used in the field of remote sensing [4,5]. The purpose of CD is to obtain land cover changes by comparing and analyzing multi-temporal remote sensing images. In this context, building change detection (BCD) has received increasing attention in the past decade and can realize effective change monitoring for single-target buildings, including new buildings and disappearing buildings. Therefore, BCD has potential value in many practical applications, such as urban development planning [6,7], evaluation of the urbanization process [8], and urban disaster prevention and mitigation [9,10].

In the past decade, many BCD methods have been proposed, and BCD has grown considerably as a result. Early on, scholars realized building extraction and change detection by designing hand-crafted features. Some studies have proposed that building recognition and change detection can be realized by using artificially designed building/shadow shape index, context, texture, and other information [11–13]. The popularization of deep

learning technology has led to remarkable results being achieved in remote sensing image interpretation [14–16]. The deep network structure has strong multi-level feature extraction capability, which can automatically learn the complex features in the remote sensing imagery, including geometric information of ground objects, spectral information, and semantic information, avoiding the reliance on a priori knowledge and the laborious process of manually constructing features. Therefore, many deep-learning-based BCD methods have been proposed and have led to some achievements. Many methods based on semantic segmentation networks have been devised to improve the performance of BCD by designing and introducing various attention mechanisms [17,18], multi-scale networks [19–21], etc. However, these networks have encountered new obstacles in further improving BCD performance. In recent years, some methods further introduced edge or contour information to enhance the performance of BCD. For instance, Bai et al. proposed an edge-guided recurrent convolutional neural network for BCD [22]. Here, the recurrent convolutional neural network has been employed for change detection [23]. The method can effectively enhance the building change extraction ability by introducing the edge of the building to guide the recurrent convolutional neural network. Yang et al. devised a multi-scale attention and edge-aware Siamese network in [24] for BCD. These methods have proved that the introduction of edge information can improve the performance of BCD. In addition, Some approaches show that the introduction of frequency domain information can effectively enhance the building outline information, thereby significantly improving the accuracy of BCD. Recently, in [25], an attention-guided high-frequency feature extraction module was proposed for BCD with the aim of enhancing the detailed feature representation of buildings. Zheng et al. proposed a high-frequency information attention-guided Siamese network for BCD. A built-in high-frequency information attention block was designed to improve the detection of building change [26]. These methods have proved that high-frequency information can effectively enhance the details such as edges, thereby further elevating the performance of BCD.

High-frequency information corresponds to the rich edge detail of ground objects in remote sensing images, and low-frequency information contains more backbone features of ground objects. Although the above-mentioned recent research shows that the introduction of high-frequency information such as edges or outlines can help improve the accuracy of BCD, the utilization of frequency by these methods is obviously insufficient. This also shows that it is potentially valuable to improve building feature representation by fully mining frequency domain information, especially considering high frequency, low frequency, and other different frequencies at the same time. To this end, we proposed a multi-scale DCT network (MDNet) for BCD. Our motivation lies in the following two aspects.

On the one hand, according to the literature [25,26], most of the current BCD methods only use the high-frequency information in the frequency domain, without considering the effect of low-frequency information and other frequency information on the representation of building features. Moreover, Reference [27] has proved that traditional global average pooling is a special case of frequency decomposition and proposed a frequency channel attention based on different frequencies to strengthen feature representation. This approach has been shown to achieve convincing performance in tasks such as image classification, object detection, and segmentation based on frequency channel attention at different frequencies. Hence, inspired by the literature [27], we can simultaneously introduce information on different frequencies in the frequency domain from the two dimensions of channel and space in order to enhance the representation ability of building features.

On the other hand, compared with traditional CD, very-high-resolution remote sensing images can distinguish finer ground targets, which makes various scales of ground objects appear at the same time, and the scene is more complicated. In view of this, the existing multi-scale methods only extract multi-scale features through multi-scale operators in the spatial domain, such as spatial pyramid pooling [28], atrous spatial pyramid pooling [29], or pyramid pooling [30]. These existing approaches lack the perception of information from the frequency domain. For this reason, introducing information on different frequencies

in multi-scale operations may help to improve the multi-scale information representation ability of existing multi-scale methods, thus leading to new means of improving the performance of BCD.

Based on the above motivations, this paper proposes a multi-scale DCT network (MDNet) for BCD. In the proposed MDNet, two novel discrete cosine transform (DCT)-based modules are utilized, namely the dual-dimension DCT attention module ($D^3AM$) and multi-scale DCT pyramid (MDP). The proposed $D^3AM$ can employ the DCT to obtain frequency information from both spatial and channel dimensions to refine building feature representation. In addition, the proposed MDP aims to enhance the multi-scale feature representation of buildings by utilizing multi-scale frequencies based on DCTs of different scales. The main contributions of this paper are as follows.

(1) We propose a novel multi-scale DCT network for BCD, which is composed of two new modules based on DCT. Our method demonstrates that using different frequencies in the frequency domain to refine features can further enhance the feature representation ability of the network.

(2) We designed a dual-dimension DCT attention module ($D^3AM$) in the proposed MD-Net, which can effectively employ frequency domain information to enhance building feature representation from both spatial and channel dimensions.

(3) Different from the existing multi-scale methods, we constructed a novel multi-scale DCT pyramid (MDP) in the proposed MDNet. The proposed MDP aims to enhance the multi-scale feature representation of buildings by utilizing multi-scale DCT.

(4) The proposed MDNet achieves a more convincing performance for three publicly available BCD datasets compared with other methods. Moreover, extensive ablation experiments also demonstrate the effectiveness of the proposed $D^3AM$ and MDP.

The remainder of this paper is arranged as follows. Section 2 summarizes some related work. Section 3 provides the proposed MDNet in detail. In Sections 4 and 5, experimental results are presented and discussed. Finally, the conclusion and future work are in Section 6.

## 2. Related Works

### 2.1. Building Change Detection in VHR Imagery

Land cover transition has been increasingly becoming an essential target for human observations of the Earth [7,31–34]. With the improvement in resolution in remote sensing, building change detection (BCD) in VHR imagery has ushered in significant challenges due to the highly complex details and the strong spatial dependence of geographic entities. Numerous advances in the research lines of building extraction and change detection have centered on the challenges associated with complex scenarios, variable scales, and small spectral interclass variance as well as large intraclass variance induced by both physical materials and illumination.

In conventional BCD methods, artificially designed building features (such as building shape index and morphological building index) are specifically designed to locally or globally extract invariant features for building instances or building changes. In [35], morphological opening and closing operators were used to extract shape-based building features with multi-scale structural elements. Zhang et al. designed a spatial feature index to measure the gray-level similarity distance in each direction in [36]. In [37], a two-stage automatic algorithm is proposed to accurately extract building features. Specifically, iteration-based morphological filtering was employed to initialize a rough result on a low-resolution model, and subsequently the combined gradient surface with the watershed algorithm was implemented as a refinement procedure. Furthermore, Bouziani et al. developed a multi-stage BCD scheme with the help of a geodatabase and prior knowledge, which can be generalized to geodatabase quality assessment and cartographic updating [7]. In order to establish a connection between the implicit building features and the attributes of morphological operations, a series of morphological building-index-based approaches has been developed. Huang et al. firstly presented the morphological building index (MBI) to realize automatic building description [11]. In addition, Huang et al. further developed

a morphological shadow index (MSI) based on the MBI, which effectively mitigates commission and omission errors in high-resolution images in combination with the MBI [12]. In [13], the variation in the MBI is considered essential for changed buildings; therefore, the building change information was decomposed in terms of the MBI, spectra, and shape dimensions. Although researchers have attempted to enhance feature engineering from multiple perspectives in BCD, hand-crafted features are limited by a priori knowledge and low-level representations.

With the rapid development of deep neural networks (DNNs), learning-based automatic feature extraction has recently been a dominant trend in computer vision [38–41]. Benefiting from powerful hierarchical structure, nonlinear transformation, and semantic representation, deep-learning-based remote sensing intelligent interpretation has attracted the interest of scholars in recent decades [42–44], especially for BCD tasks. In [45], Li et al., employed a residual U-Net network to distinguish building changes from a fused difference image. For the integration of multi-temporal remote sensing features, a multi-branch network structure was specifically designed to fuse semantic information about building changes at different levels [46]. The Siamese structure is gradually being widely applied as a basic backbone in BCD for multi-temporal analysis [47]. Liu et al. designed a deep Siamese convolutional network to simultaneously perform change detection and semantic segmentation in [48], which enables significant enhancement of feature distinguishability. In [49], the self-attention module was utilized to fully exploit the spatiotemporal correlation on the basis of a Siamese network. In [22], with the multilevel features obtained by the Siamese network, long short-term memory and edge prior augmentation were introduced to further analyze the difference information, which contributes to enhancing the building boundaries. Similarly concerned with accurate segmentation of edges, i.e., high-frequency information, in dense buildings, Chen et al. further exploited edge information to guide hierarchical transformers for long-term contextual modeling and feature refinement [50]. Even though high-frequency information is beneficial to BCD performance [26], we provide insight distinctly showing that it is insufficient to solely focus on high-frequency information, and in this study we leveraged information at different frequencies to strengthen the feature representation.

### 2.2. Multi-Scale Feature Learning in Semantic Segmentation

BCD in VHR imagery is capable of distinguishing finer ground targets compared to traditional change detection, which results in the simultaneous appearance of features at various scales and more complex scenarios [51]. Multi-scale feature learning, which has achieved great success in natural image analysis, is considered as a potential solution to the above problems. Atrous spatial pyramid pooling (ASPP), proposed in [52], employs concatenated atrous convolutional layers with different expansion rates for capturing multi-scale information. The pyramid pooling module in pyramid scene parsing network (PSPNet), proposed by Zhao et al., is designed to collect hierarchical multi-scale information, which is more representative than global pooling [30]. Instead of complex dilated convolution and artificial decoder networks, Li et al. combine an attention mechanism and a spatial pyramid to provide better features for dense prediction [53]. Li et al. present DFANet, which performs multi-scale feature propagation via subnetwork aggregation and substage aggregation [54]. It reduces the number of parameters while obtaining a sufficient receptive field to enhance the representation of the model. The attention-based hierarchical multi-scale prediction method is proposed for semantic segmentation [55], enabling the network to support specific scales that specialize in prediction.

Inspired by multi-scale feature fusion and reuse techniques in semantic segmentation, multi-scale information is still fully exploited in remote sensing. An end-to-end multi-scale adaptive feature fusion network was proposed in [56] to adaptively mitigate the problem of large differences in object sizes. In addition, Liu et al. propose a local–global pyramid network (LGPNet) to extract discriminative building features with variable scales from both global and local dimensions [57]. Zhang et al. further designed a deep multi-scale

multi-attention Siamese transformer network [58]. The fusion of different stages of features in the Siamese feature extractor within the multi-attention Siamese transformer architecture allows the detection of targets with different degrees of irregularity. Existing multi-scale feature learning techniques are more inclined to extract multi-scale information through the spatial domain; nevertheless, this paper aims to strengthen the multi-scale representation capability of the network from the frequency perspective instead.

### 2.3. Attention Mechanism

Inspired by the human visual system, i.e., the ability to locate salient regions in complex scenes naturally and efficiently, the attention mechanism has been introduced into computer vision. It can adaptively recalibrate the weight assigned to elements of interest via the input image features [59–61].

Channel attention aims at adaptive selection of channel information. Hu et al. proposed the squeeze-and-excitation (SE) block to capture the channel relationships and enhance representation [39]. Subsequently, various improved versions of the SE block were developed. Instead of simply utilizing global average pooling to collect global information, the global second-order pooling (GSoP) model was designed to construct higher-order statistical information [62]. Yang et al. replaced the original SE block with gated channel transformation to reduce the computation and parameter cost associated with the fully connected layer [63].

In order to make convolutional neural networks more attentive to key regions with transform-invariant properties, Jaderberg et al. proposed spatial transformer networks to explicitly learn transform invariance [64]. Spatial attention can be considered as a form of spatial information localization. Furthermore, Gong et al. proposed the Gaussian spatial attention module for the hyperspectral image change detection task, which adaptively constructs a spatial Gaussian distribution and samples each image patch to focus on the spatial region relevant to the center pixel [65].

In BCD, the attention mechanism is also extensively applied to enhance the critical instances which contribute to building extraction and difference information analysis. In [20], a pyramid feature-based attention-guided Siamese network is proposed to stress the correlation between input pairs and enhance the long-distance dependency. Song et al. enhanced the discrimination of building change features with the help of both spatial attention and channel attention, eventually formulating the attention-guided end-to-end change detection network [18]. In [66], an ensemble channel attention module is proposed by Fang et al. to refine the semantic features at different levels. In addition, a high-frequency attention-guided Siamese network (HFA-Net) is proposed in [26] to utilize the attention mechanism for addressing the poor segmentation of building boundaries. In contrast, in this study, we attempted to mine frequency information from both the spatial dimension and the channel dimension to refine the building feature map.

## 3. Methodology

To better detect fine-grained multi-scale land cover objects, a new deep neural network, the multi-scale DCT network (MDNet), is proposed in this work. Different from conventional deep neural networks for CD, it has two novel modules, i.e., the dual-dimension DCT attention module ($D^3AM$) and multi-scale DCT pyramid (MDP), to better dig and utilize the information from frequency domain for better CD performance. The $D^3AM$ employs the DCT to concurrently obtain frequency information from both spatial and channel dimensions to refine the feature maps in the network. Furthermore, the MDP can better recognize the objects with varied sizes through the information obtained by multi-scale DCT.

In this section, the overview of MDNet will be given briefly in Section 3.1. Then, the $D^3AM$ and MDP will be introduced in detail in Sections 3.2 and 3.3, respectively.

### 3.1. Overview

We employed a U-shaped [67] network with skip connections as the backbone of the proposed MDNet as shown in Figure 1. This backbone architecture can automatically acquire multi-scale feature representation and obtain hierarchical feature maps with varied sizes, which can be helpful for acquiring better cognition of multi-scale land cover objects in BCD. With these extracted multi-scale feature maps, we propose the $D^3AM$ to obtain extra information from the transformation domain, which is used to revalue the significance of feature maps from both spatial and channel dimensions for finer feature representation. We used the MDP to further dig multi-scale frequency information and built a feature pyramid through multi-scale DCT, which can acquire better recognition of multi-scale land cover objects. Apart from these modules, we used two conventional blocks, which contain several convolutional layers with batch normalization and rectified linear unit (ReLU) activation functions, to preliminarily process the input bi-temporal remote sensing images and generate the binary change maps, respectively. Moreover, to create hierarchical multi-scale features, down-sampling and up-sampling layers are utilized in MDNet. These layers have pooling layers and bi-linear interpolation with convolutional layers, respectively.

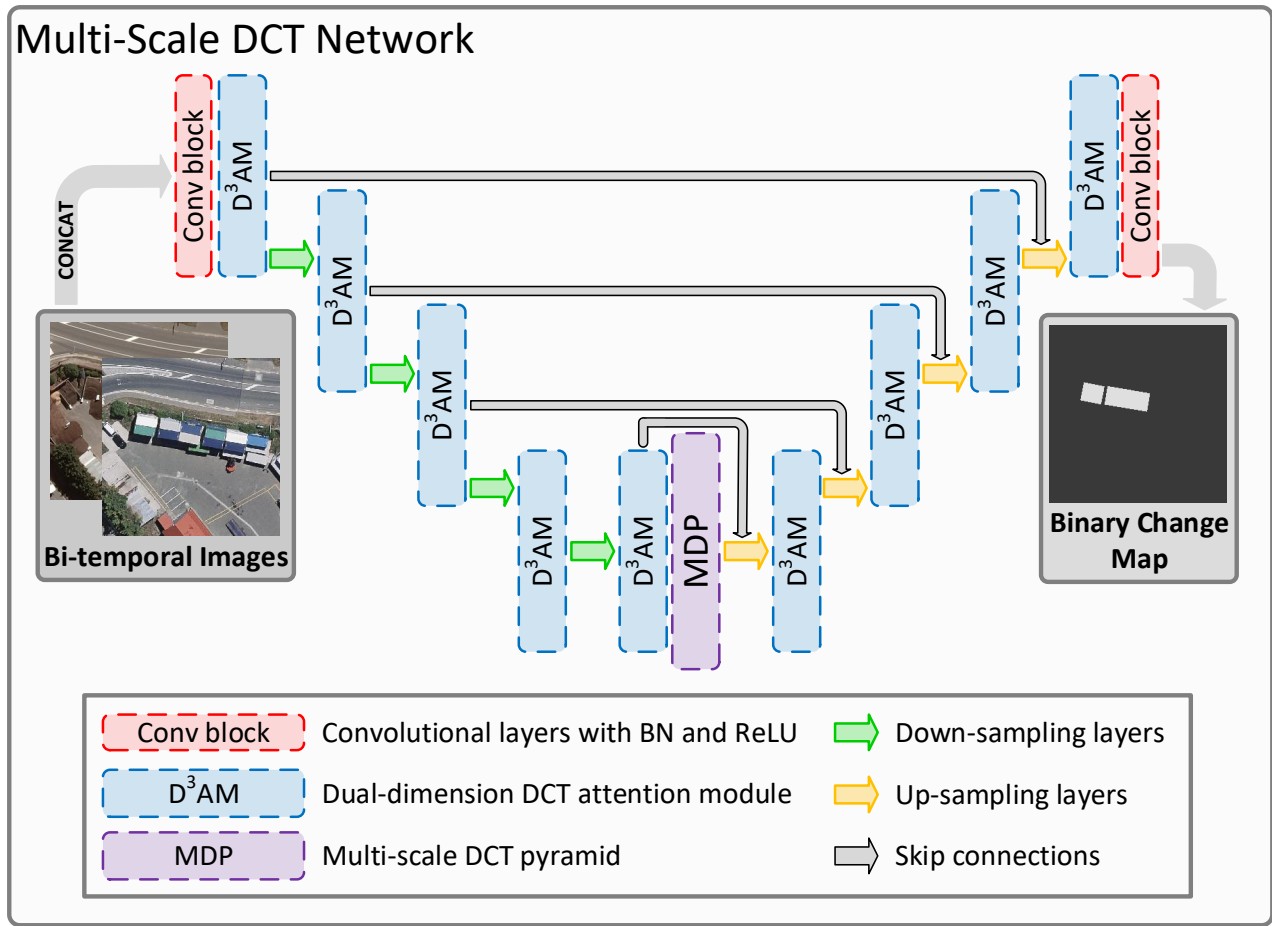

**Figure 1.** A brief graphical demonstration of the proposed MDNet.

The proposed MDNet conducts the BCD in VHR bi-temporal remote sensing images as follows: Firstly, the bi-temporal remote sensing images are concatenated in the channel dimension and input into MDNet. Then, multi-scale features are extracted by up-sampling and down-sampling layers. These features are refined and enhanced by the $D^3AM$ and MDP. When features are processed by up-sampling layers, skip connections allow the features from the early stage to participate in the generation of the binary change maps, which provide more information to improve CD performance [26]. Finally, the binary

change maps are produced and output to indicate the change in land cover objects with pixel-wise results.

### 3.2. Dual-Dimension DCT Attention Module

In [26], it has been demonstrated that frequency domain information can improve the performance of BCD. The researchers found that enhancing the high-frequency information can help the network better detect land cover objects with clearer boundaries, which improve CD performance. However, it is well recognized that most noise is concentrated in the high-frequency region. Individually enhancing the high-frequency information may enhance noise and cause performance loss for CD. High-frequency information is not the only significant aspect for image recognition [27]; ignoring information from the lower frequency domain when building the attention mechanism can potentially decrease the BCD performance in complex scenes. Based on this, we utilize DCT to obtain more complete frequency information from both spatial and channel dimensions in the proposed $D^3AM$. With the dual-dimension frequency information acquired, spatial and channel attention scores are generated to enhance useful information in the feature maps for better BCD performance. The process is illustrated in Figure 2 and as follows:

Let the input of the $D^3AM$ be represented as $\mathcal{F}^{in} \in \mathbb{R}^{C \times H \times W}$, where $H, W, C$ indicate the height, weight, and channel size, respectively. To acquire the information from both spatial and channel dimensions, the input feature maps $\mathcal{F}^{in}$ are input into two different DCT attention paths concurrently, i.e., the (a) spatial DCT branch and (b) channel DCT branch.

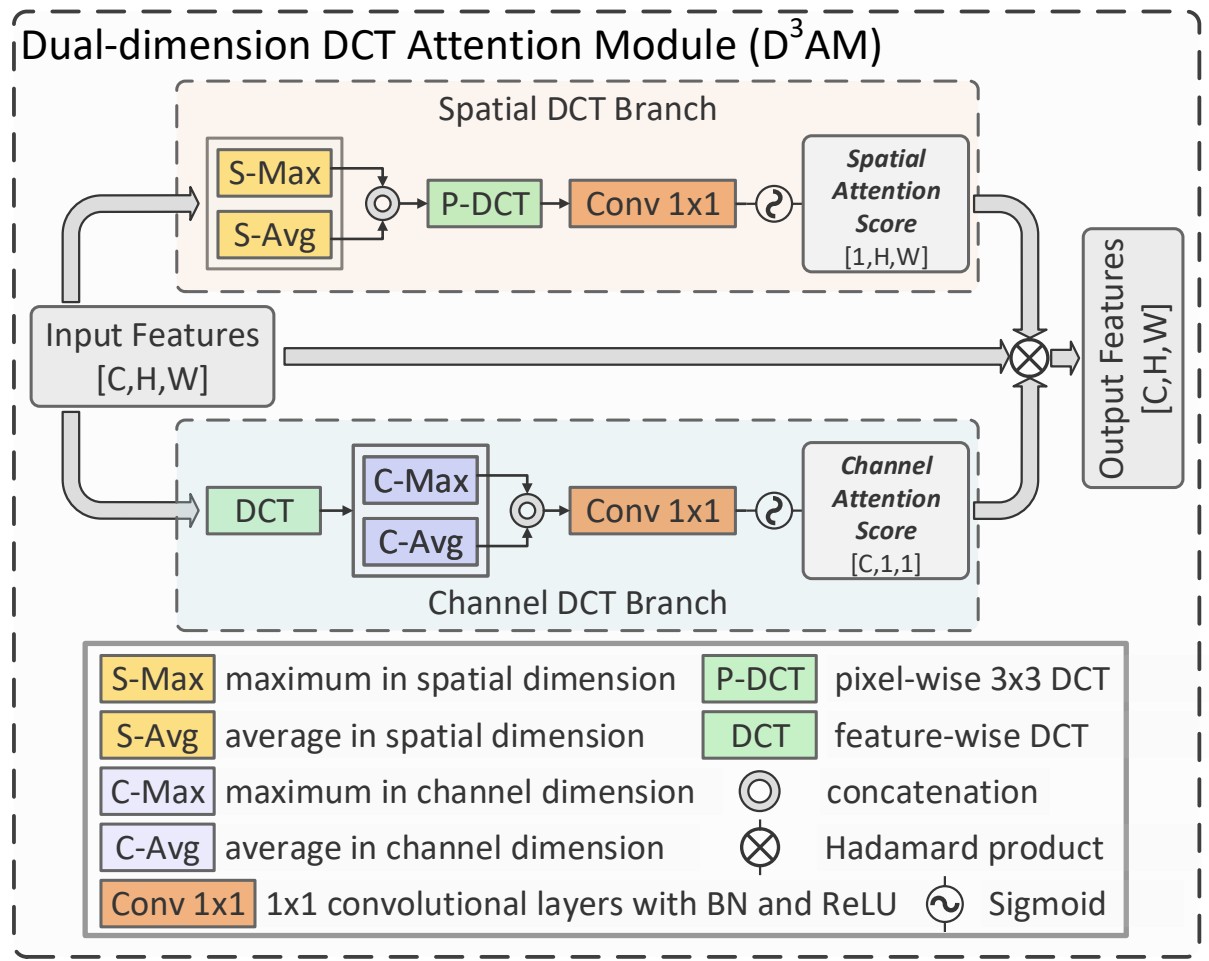

**Figure 2.** A brief graphical demonstration of the proposed $D^3AM$. A residual link is utilized to keep the output stable and facilitate supervised learning.

### 3.2.1. (a) Spatial DCT Branch

In this path, the most representative spatial features are firstly extracted by the maximum and average in the spatial dimension [68] and concatenated as $\mathcal{F}^{s1} \in \mathbb{R}^{2 \times H \times W}$, which can be represented as follows:

$$\mathcal{F}^{s1} = Concat\left\{ Max_S\left(\mathcal{F}^{in}\right), Avg_S\left(\mathcal{F}^{in}\right)\right\} \tag{1}$$

where $Max_S(\cdot)$ and $Avg_S(\cdot)$ indicate the procedures of taking the maximum and average in the spatial dimension, respectively. Furthermore, $Concat\{\cdot\}$ represents the concatenation in the channel dimension. Then, we employ a pixel-wise DCT with a kernel size of $3 \times 3$ to acquire the fine-grained spatial frequency information $\mathcal{F}^{s1d} \in \mathbb{R}^{2 \times H \times W}$, which can be represented as:

$$\mathcal{F}^{s1d} = DCT_P^{3 \times 3}\left(\mathcal{F}^{s1}\right) \tag{2}$$

where $DCT_P^{3 \times 3}(\cdot)$ is a pixel-wise DCT with a kernel size of $3 \times 3$. At the end of spatial DCT branch, the spatial attention mask $\mathcal{A}^s \in \mathbb{R}^{1 \times H \times W}$ is obtained by several convolutional layers with a kernel size of $1 \times 1$ and a sigmoid function, which can be illustrated as:

$$\mathcal{A}^s = \delta\left(Conv^s\left(\mathcal{F}^{s1d}\right)\right) \tag{3}$$

where $Conv^s(\cdot)$ indicates several $1 \times 1$ convolutional layers with BN and the ReLU, which are employed to evaluate the features in the spatial dimension through supervised learning. Furthermore, a sigmoid function, $\delta(\cdot)$, is employed to obtain a stable attention score.

### 3.2.2. (b) Channel DCT Branch

Different from the spatial branch, we firstly use a feature-wise DCT to acquire the channel-wise global frequency information $\mathcal{F}^{c1d} \in \mathbb{R}^{C \times H \times W}$, which can be illustrated as:

$$\mathcal{F}^{c1d} = DCT_f^{H \times W}\left(\mathcal{F}^{in}\right). \tag{4}$$

To acquire the most representative channel-wise frequency information [68] $\mathcal{F}^{c1} \in \mathbb{R}^{2C \times 1 \times 1}$, the global max pooling and global average pooling are employed, which can be represented as:

$$\mathcal{F}^{c1} = Concat\left\{ Max_C\left(\mathcal{F}^{c1d}\right), Avg_C\left(\mathcal{F}^{c1d}\right)\right\} \tag{5}$$

where $Max_C(\cdot)$ and $Avg_C(\cdot)$ represent the global max pooling and global average pooling, respectively. Then, we train several $1 \times 1$ convolutional layers to generate channel-wise attention score $\mathcal{A}^c \in \mathbb{R}^{C \times 1 \times 1}$, which can be illustrated as follows:

$$\mathcal{A}^c = \delta\left(Conv^c\left(\mathcal{F}^{c1}\right)\right) \tag{6}$$

where $Conv^c(\cdot)$ indicates several convolutional layers with BN, the ReLU, and a kernel size of $1 \times 1$. Notably, a sigmoid function is also employed to acquire a stable output in this branch.

Through the spatial DCT branch and channel DCT branch, the dual-dimensional attention scores $\mathcal{A}^s$ and $\mathcal{A}^c$ are obtained. Then, the final output of the D$^3$AM, $\mathcal{F}^{out} \in \mathbb{R}^{C \times H \times W}$, can be obtained as follows:

$$\mathcal{F}^{out} = \mathcal{F}^{in} + \mathcal{A}^s \otimes \mathcal{F}^{in} \otimes \mathcal{A}^c \tag{7}$$

where $\otimes$ indicates the Hadamard product. As shown in the equation, a residual link is utilized at this stage to keep the output stable and facilitate supervised learning.

In sum, in the proposed D$^3$AM, spatial and channel-wise frequency information can be exploited by using dual-dimensional DCT. Then the attention mechanisms are built

around these information resources and refine the feature representation from both spatial and channel dimensions, which make the proposed $D^3AM$ different from conventional frequency-analysis-based attention mechanisms.

*3.3. Multi-Scale DCT Pyramid*

In the proposed MDNet, multi-scale feature representation is acquired not only from the hierarchical backbone but also in the proposed MDP. Multi-scale feature extraction has proven useful for obtaining finer pixel-wise annotation for CD tasks [21]. Basically, the deepest feature maps have smaller spatial size compared to other features, thus making them lack precise spatial information. Based on this fact, it can be helpful to extract multi-scale features based on these deep features and improve the recognition of multi-scale land cover objects. Many conventional CD methods adopt adaptive pooling and atrous convolutional layers to extract multi-scale features for better cognition of varied land cover objects [21], which is revealed to be helpful. However, these directly extracted features can be rough, since they are built over the features with minor spatial information. To overcome this problem, we utilized the spatial frequency analysis to refine the spatial information of multi-scale features in the proposed MDP, as shown in Figure 3. Inspired by [52], we firstly used dilated convolutional layers to build multi-scale features with different degrees of spatial detail information. Then, we used DCT with different scales to refine these multi-scale features. With finer multi-scale feature maps, the land cover objects with varied sizes can be better recognized and detected in the proposed MDNet. The detailed procedure of the proposed MDP is demonstrated as follows:

Firstly, let $\mathcal{P}^{in} \in \mathbb{R}^{C \times H \times W}$ be the input of the MDP. Then, the multi-scale feature maps $\mathcal{P}_x \in \mathbb{R}^{C \times H \times W}\ \{x = 1, 2, 3, 4, 5\}$ can be obtained as follows:

$$
\begin{aligned}
\mathcal{P}_1 &= Conv_P^1\left(\mathcal{P}^{in}\right) \\
\mathcal{P}_2 &= DilatedConv_P^6\left(\mathcal{P}^{in}\right) \\
\mathcal{P}_3 &= DilatedConv_P^{12}\left(\mathcal{P}^{in}\right) \\
\mathcal{P}_4 &= DilatedConv_P^{18}\left(\mathcal{P}^{in}\right) \\
\mathcal{P}_5 &= Up\left(Conv_P^1\left(GAP\left(\mathcal{P}^{in}\right)\right)\right)
\end{aligned}
\tag{8}
$$

where $Conv_P^1(\cdot)$ denotes a convolutional layer with a kernel size of $1 \times 1$, BN, and the ReLU. $DilatedConv_P^n(\cdot)$ represents the $3 \times 3$ dilated convolution with the dilated rate of n. Furthermore, $GAP(\cdot)$ indicates global average pooling. Then, we use a multi-scale DCT process to extract multi-scale frequency information to refine these features as shown in Figure 3, which can be denoted as:

$$
\mathcal{P}_x^d = MS - DCT^{N \times N}(\mathcal{P}_x)
\tag{9}
$$

where $MS - DCT^{N \times N}(\cdot)$ indicates the multi-scale DCT with a kernel size of N $\times$ N $\{N = 3, 5, 7, 9, 11\}$. Then, the spatially enhanced multi-scale features are fused to generate the output, $\mathcal{P}^{out} \in \mathbb{R}^{C \times H \times W}$, as follows:

$$
\mathcal{P}^{out} = Conv_P^1\left(Concat\left\{\mathcal{P}_1^d, \cdots, \mathcal{P}_5^d\right\}\right)
\tag{10}
$$

To sum up, the proposed MDP conducts frequency-based analysis over multi-scale feature maps to refine them in the spatial dimension, which can enhance their spatial information for better recognition of multi-scale land cover objects in remote sensing imagery.

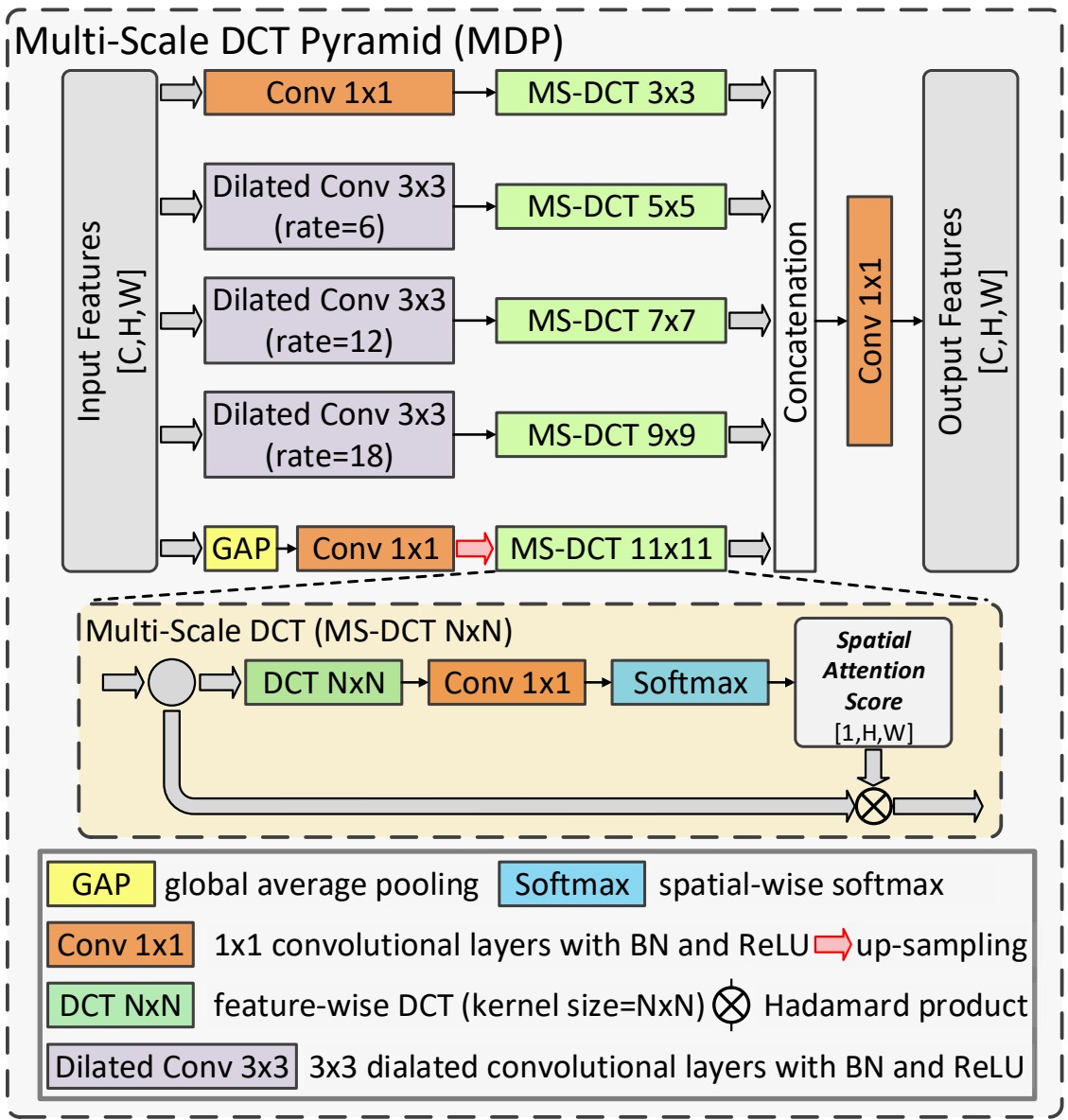

**Figure 3.** A brief graphical demonstration of the proposed MDP. A residual link is utilized in the MS-DCT.

### 4. Experimental Results

*4.1. Datasets*

To validate the effectiveness of our proposed method, we conducted experiments on three benchmark remote sensing change detection datasets, namely the WHU-CD Dataset [69], LEVIR-CD Dataset [49] and Google Dataset [70]. Below are the detailed descriptions of these three datasets.

#### 4.1.1. WHU-CD Dataset

The WHU-CD Dataset was released by Wuhan University in 2019. This dataset is a publicly available collection used for remote sensing image processing and building recognition, widely applied in research and development work within the fields of change detection.

In the WHU-CD Dataset, buildings exhibit variations in color, shape, and style, including both missing and reconstructed structures. Additionally, the entire remote sensing scene encompasses complex grasslands, bridges, trees, and other ground objects, which poses significant challenges for accurate building detection. This dataset comprises aerial images

captured at the same location in Christchurch, New Zealand, in two periods, 2012 and 2016. The images depict the reconstruction situation of the area after a 6.3-magnitude earthquake occurred in February 2011. These images possess a high resolution, with dimensions of 32,507 × 15,354 pixels and a spatial resolution of 0.075 m per pixel. The coverage area reaches up to 20.5 square kilometers. The specific partitioning method of the dataset is consistent with the approach described in [26,49,57,66,71]. Specifically, the complete WHU-CD Dataset was divided into 3480 pairs of image pairs, each sized 256 × 256 pixels. Out of these pairs, 2396 were allocated for training purposes, while the remaining 1084 pairs were used for testing.

### 4.1.2. LEVIR-CD Dataset

The LEVIR-CD Dataset covers various types of buildings, including villa residences, high-rise apartments, small garages, and large warehouses. We focused on changes related to buildings, including the growth of buildings from land/grass/paved areas to newly constructed building regions and the decay of buildings. This dataset consists of 637 Google Earth image patches, with a resolution of 0.5 m per pixel, and each patch is sized at 1024 × 1024 pixels. These images were extracted from dual-temporal imagery spanning a period of 5 to 14 years, showcasing evident land use changes, particularly in building growth. The dual-temporal images have been annotated by remote sensing image interpretation experts using binary labels, where 1 indicates change and 0 indicates no change. Each sample is annotated by one annotator and subsequently double-checked by another annotator to ensure annotation quality. In total, the LEVIR-CD Dataset comprises 31,333 independent instances of changed buildings. In the experiment, consistent with [26,49,57,66,71], we cropped the LEVIR-CD Dataset into 10,192 pairs of images, each sized at 256 × 256 pixels. The training set and test set consist of 7120 pairs and 3072 pairs of images, respectively.

### 4.1.3. Google Dataset

The Google Dataset contains diverse and complex buildings, partly due to the large-scale terrain changes brought about by urbanization. These changes pose challenges in accurately extracting buildings within complex scenes such as trees, roads, and lakes. In complex scenes, building extraction faces multiple issues. For instance, tree shadows may obscure parts of the buildings, roads and other objects may overlap with buildings, and bodies of water like lakes may reflect the imagery of the buildings. Moreover, there are also problems with occlusion, lighting variations, and remote sensing image quality, further increasing the difficulty of accurately extracting buildings from complex scenes. The Google Dataset, released by Peng et al. [70] in 2021, primarily focuses on urban changes in the suburban area of Guangzhou, China, from 2006 to 2019. The dataset consists of 19 pairs of very-high-resolution (VHR) images captured in different seasons, with sizes ranging from 1006 × 1168 to 4936 × 5224 pixels and a resolution of 0.55 m. Consistent with [26,49,57,66,71], we divided the Google Dataset into 3130 pairs of images with 256 × 256 pixels, where 2191 pairs for training and 939 pairs for testing purposes.

### 4.2. Evaluation Metrics

To comprehensively evaluate the performance of change detection algorithms, four commonly used metrics, namely precision, recall, F1-score, and intersection over union (IOU), are used to report the quantitative results analysis. We use *TP*, *TN*, *FP*, and *FN* to represent the number of true positives, true negatives, false positives, and false negatives, respectively. The detailed definitions of these four evaluation metrics are as follows. Precision measures the proportion of true positives among the samples predicted as positive, which can be represented by the following formula:

$$Precision = \frac{TP}{TP + FP} \tag{11}$$

A higher precision indicates a stronger ability of the classifier or model to correctly identify true positives among the samples predicted as positive.

Recall measures the proportion of true positives among the actual positive samples, which can be represented by the following formula:

$$Recall = \frac{TP}{TP + FN} \tag{12}$$

A higher recall indicates a stronger ability of the classifier or model to correctly identify true positives among the actual positive samples.

F1-score is a comprehensive evaluation metric that balances precision and recall, which can be represented by the following formula:

$$F1\text{-}Score = \frac{2 \times Recall \times Precision}{Recall + Precision} \tag{13}$$

The F1-score considers both precision and recall, providing a representative measure for evaluating the overall performance of a classifier or model.

IOU is primarily used in object detection tasks to measure the overlap between the predicted results and the ground truth objects. It can be calculated by dividing the intersection area of the predicted box (or region) and the ground truth box by the union area of the two as follows:

$$IOU = \frac{TP}{TP + FN + FP} \tag{14}$$

A higher IOU value indicates that the predicted segmentation result is closer to the ground truth segmentation target, thereby indicating better performance of the model.

### 4.3. Implementation Details

To verify the effectiveness of our proposed method, eight excellent peers were selected as comparative algorithms, namely FC-EF [47], FC-Siam-Diff [47], FC-Siam-Conc [47], STANet [49], SNUNet [66], BIT [71], and LGPNet [57]. These comparison methods have been detailed in the related work section. To ensure the fairness of the experiments, all the comparative methods were reproduced with their respective source codes, and the reported results were based on their optimal parameters.

In the experiments, MDNet was executed on the PyTorch platform with CUDA 11.3, utilizing a single NVIDIA RTX 3090 GPU with 24 GB video memory. In the hyperparameter configuration, a batch size of 8 was used, and stochastic gradient descent (SGD) was chosen as the optimizer. The initial learning rate was set to 0.01, with a momentum of 0.9 and a weight decay rate of $10^{-5}$. The learning rate decayed at the 10th, 15th, 30th, and 40th epochs by a factor of 0.1.

### 4.4. Comparison with Other Methods

4.4.1. Results for the WHU-CD Dataset

Table 1 provides a quantitative analysis of accuracy, recall, F1-score, and IOU metrics for the WHU-CD Dataset. We highlight the best results in the table by using bold font. It can be observed from the experimental results that MDNet achieved the best results in three out of the four metrics, except for a slightly lower performance in the recall metric compared to the top-performing algorithm (FC-Siam-Diff). In more detail, MDNet achieved a performance of 91.456% in F1-score and 84.257% in IOU on these two comprehensive metrics, surpassing the second-best algorithm (LGPNet) by 4.874% and 1.129%, respectively. It is worth noting that such a significant lead in building change detection on VHR images is truly remarkable.

Figure 4 shows a comparison of some representative visual results for the WHU-CD Dataset. The visual results demonstrate that MDNet has achieved outstanding performance compared to other comparative methods, whether detecting large-scale building changes or subtle ones. The misidentification issue in the FC-Siam-Diff, FC-EF, and FC-Siam-

Conc methods is severe. This is because these three models lack the ability to effectively discriminate against backgrounds that are similar to buildings, leading to a higher chance of mistakenly identifying objects with low inter-class similarity as buildings. LGPNet and STANet also face the aforementioned issue with the scenes of large or dense buildings. However, these two algorithms handle the problem of missing detection relatively well. The most severe issue of missing detection was found with SNUNet. SNUNet struggles with detecting even clearly outlined buildings, let alone small-scale building changes. The problem of false negatives in SNUNet is quite pronounced and cannot be overlooked. BIT performs decently overall, but it falls short compared to MDNet in terms of fine-grained details. MDNet achieves excellent results both in capturing the basic outlines of large buildings and in detecting the edges of small-scale structures.

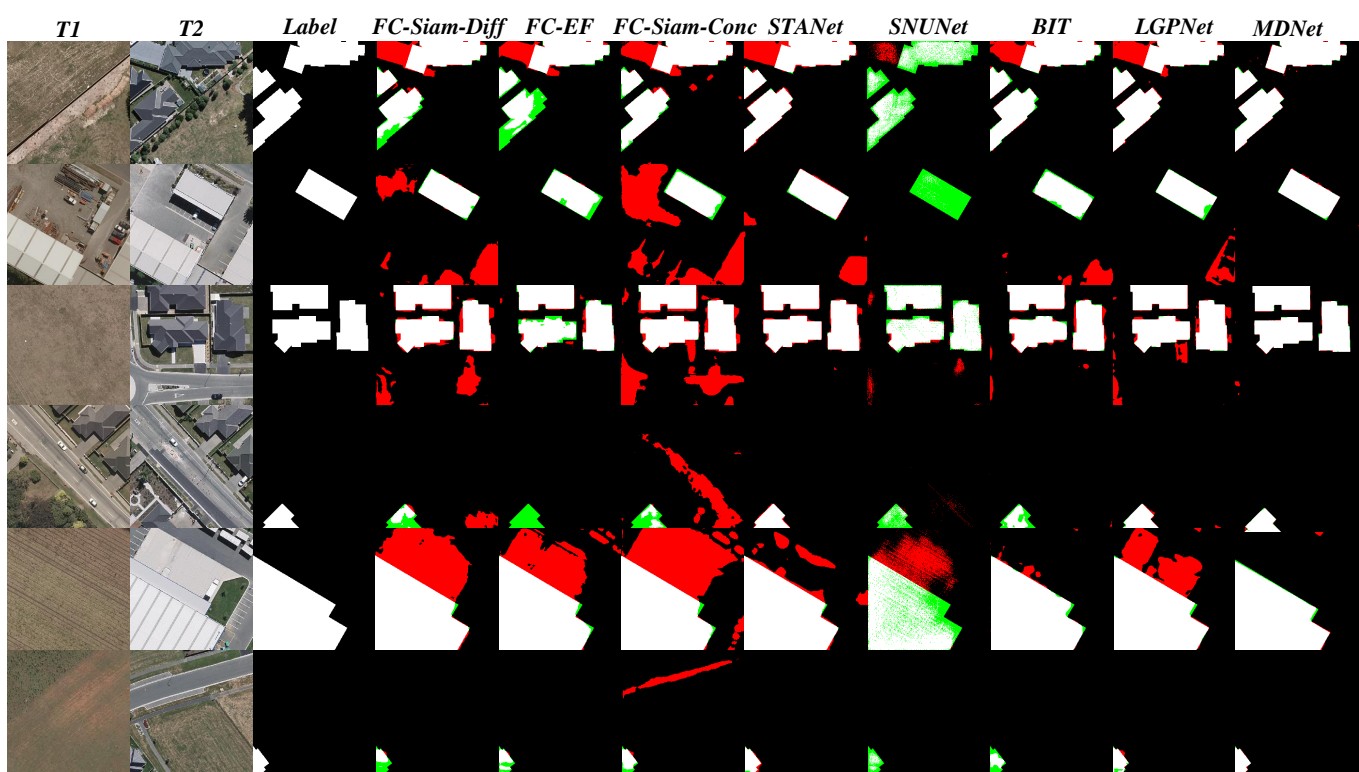

**Figure 4.** Some representative visualization results for the WHU-CD Dataset. Each column, from left to right, represents the T1 image, T2 image, ground truth, FC-Siam-Diff, FC-EF, FC-Siam-Conc, STANet, SNUNet, BIT, LGPNet, and our proposed MDNet. Green and red represent missing and false detection pixels, while white and black represent correctly detected changed and unchanged pixels, respectively.

**Table 1.** Quantitative experimental results (in %) of precision, recall, F1-score, and IOU for different change detection methods for the WHU-CD Dataset. The best results are highlighted in bold.

| Methods | Precision | Recall | F1-Score | IOU |
|---|---|---|---|---|
| FC-Siam-Diff [47] | 67.116 | **88.850** | 76.468 | 61.902 |
| FC-EF [47] | 90.134 | 79.473 | 84.468 | 73.112 |
| FC-Siam-Conc [47] | 50.710 | 88.402 | 64.490 | 47.546 |
| STANet [49] | 82.268 | 88.552 | 85.294 | 74.360 |
| SNUNet [66] | 88.436 | 62.294 | 73.098 | 57.602 |

**Table 1.** *Cont.*

| Methods | Precision | Recall | F1-Score | IOU |
|---|---|---|---|---|
| BIT [71] | 83.094 | 86.350 | 84.691 | 73.446 |
| LGPNet [57] | 84.705 | 88.544 | 86.582 | 76.340 |
| **MDNet (Ours)** | **94.678** | 88.445 | **91.456** | **84.257** |

4.4.2. Results for the LEVIR-CD Dataset

Table 2 presents the qualitative analysis results of MDNet and comparative methods in building change detection in four scenarios for the LEVIR-CD Dataset. From the results in Table 2, it can be observed that MDNet continued its outstanding performance, except for not achieving the best accuracy. State-of-the-art results were achieved for recall along with the two comprehensive metrics, F1-score and IOU. In the two comprehensive metrics that require special attention, MDNet outperformed the second-ranked algorithms, LGPNet 0.678% and 1.129%, in terms of F1-score and IOU, respectively.

**Table 2.** Quantitative experimental results (in %) of precision, recall, F1-score, and IOU for different change detection methods for the LEVIR-CD Dataset. The best results are highlighted in bold.

| Methods | Precision | Recall | F1-Score | IOU |
|---|---|---|---|---|
| FC-Siam-Diff [47] | **94.001** | 77.330 | 84.854 | 73.693 |
| FC-EF [47] | 87.481 | 80.501 | 83.846 | 72.186 |
| FC-Siam-Conc [47] | 92.698 | 78.366 | 84.931 | 73.809 |
| STANet [49] | 85.468 | 83.786 | 84.618 | 73.338 |
| SNUNet [66] | 91.223 | 82.921 | 86.874 | 76.794 |
| BIT [71] | 90.551 | 86.454 | 88.455 | 79.300 |
| LGPNet [57] | 92.323 | 87.806 | 90.008 | 81.831 |
| **MDNet (Ours)** | 91.401 | **89.982** | **90.686** | **82.960** |

In addition, to further validate the effectiveness of MDNet for the LEVIR-CD Dataset, we provide some representative visual case comparisons in Figure 5. It can be observed from the figure that the LEVIR-CD Dataset is characterized by highly similar variations in buildings and backgrounds, which poses a significant challenge to the model's generalization ability. However, MDNet surpasses its peers in terms of overall change detection performance, demonstrating significantly fewer misidentifications and omissions. In more detail, taking the examples in the first and third rows, the changing buildings occupy a very small portion of the overall scene. This tests the model's detection ability for architectural objects at different scales. MDNet demonstrates excellent performance in extracting the edges of the entire small building contours. Regarding misidentifications, these algorithms perform well, mainly because the proportion of buildings in the T1 moment of the LEVIR-CD Dataset is very low, making it less prone to misidentifications. However, in terms of omissions, the compared algorithms in the same field perform worse than our proposed algorithm in different scenarios. Specifically, FC-Siam-Diff, FC-EF, and FC-Siam-Conc algorithms show poor performance in dense building areas, while other algorithms have slight flaws in detecting buildings that are too similar to the background.

4.4.3. Results for the Google Dataset

Table 3 presents the results of MDNet and peer comparators for four metrics in the Google Dataset. MDNet ranked first in accuracy, F1-score, and IOU and second for recall. To be more specific, in the two key comprehensive metrics, F1-score and IOU, MDNet outperformed the second-ranked algorithm LGPNet by 0.982% and 1.452%, respectively.

MDNet achieved highly competitive results for building change detection datasets that were collected from three different locations, exhibit different architectural styles, and vary in data scale. This demonstrates that both at the module and overall framework levels, the design of MDNet exhibits strong generalization capabilities.

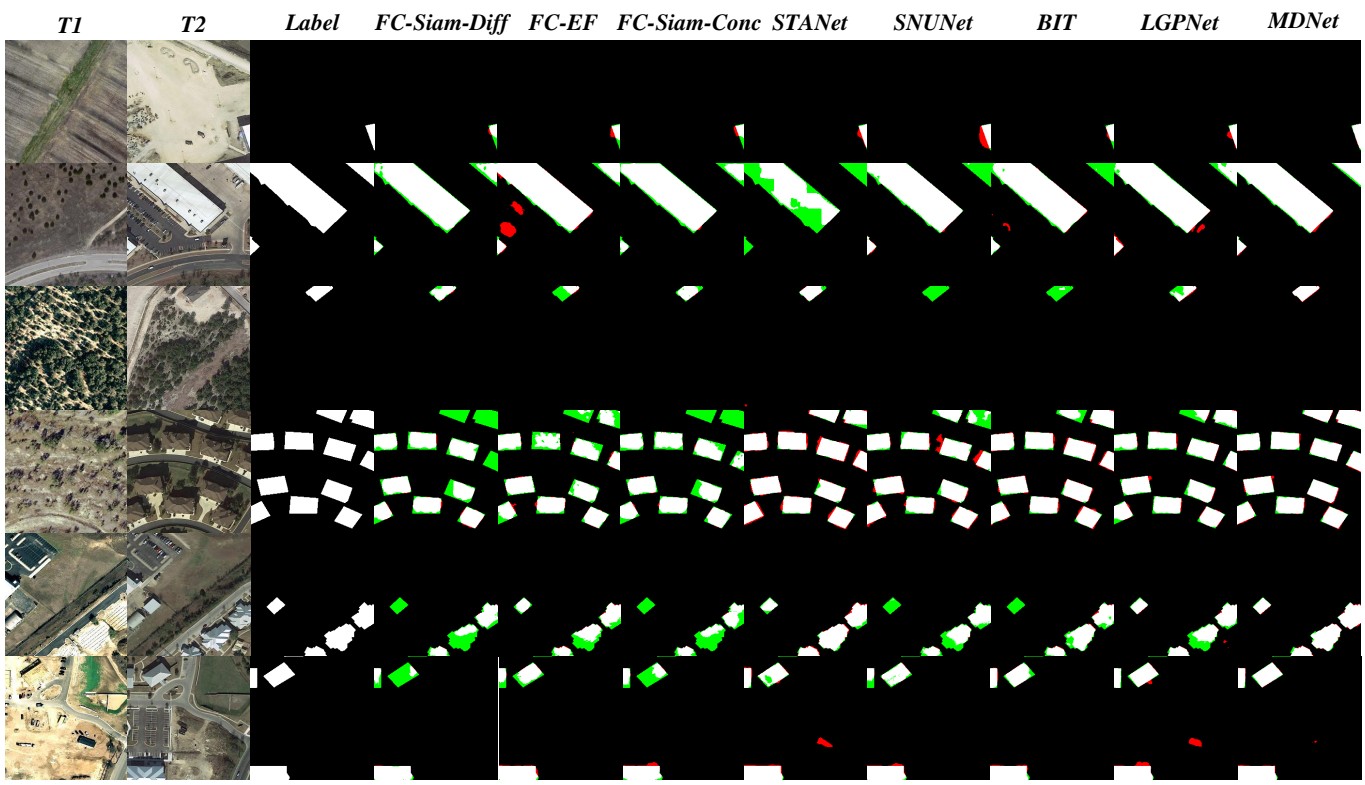

**Figure 5.** Some representative visualization results for the LEVIR-CD Dataset. Each column, from left to right, represents the T1 image, T2 image, ground truth, FC-Siam-Diff, FC-EF, FC-Siam-Conc, STANet, SNUNet, BIT, LGPNet, and our proposed MDNet. Green and red represent missing and false detection pixels, while white and black represent correctly detected changed and unchanged pixels, respectively.

**Table 3.** Quantitative experimental results (in %) of precision, recall, F1-score, and IOU for different change detection methods for the Google Dataset. The best results are highlighted in bold.

| Methods | Precision | Recall | F1-Score | IOU |
|---|---|---|---|---|
| FC-Siam-Diff [47] | 80.525 | 41.560 | 54.825 | 37.764 |
| FC-EF [47] | 84.087 | 55.141 | 66.605 | 49.931 |
| FC-Siam-Conc [47] | 81.876 | 54.643 | 65.543 | 48.747 |
| STANet [49] | 80.231 | 52.434 | 63.420 | 46.434 |
| SNUNet [66] | 55.669 | 47.705 | 51.381 | 34.572 |
| BIT [71] | 89.623 | 68.641 | 77.741 | 63.587 |
| LGPNet [57] | 83.917 | **82.467** | 83.186 | 71.212 |
| **MDNet (Ours)** | **88.356** | 80.359 | **84.168** | **72.664** |

Furthermore, to further illustrate the superior performance of MDNet for the Google Dataset in a more comprehensive manner, we present some representative visual results in Figure 6. From a general perspective, it can be concluded from the figure that compared to the previous two benchmark datasets WHU-CD and LEVIR-CD, the Google Dataset is

significantly more challenging, as it consists of large-scale scene changes and fine-grained building changes within a small area. In more detail, the five algorithms, FC-Siam-Diff, FC-EF, FC-Siam-Conc, STANet, and SNUNet, exhibit significant issues with false negatives in detection. This can be observed in the second, fourth, and fifth rows of the figure, where these models lack the ability to accurately identify buildings that have high similarity to the background. On the other hand, BIT and LGPNet perform less effectively than MDNet in controlling false positives. MDNet actually achieves good overall boundary control. However, its only flaw is in the detection of shadows on buildings, where MDNet may produce false positive detections. Overall, MDNet is a remarkable algorithm with its flaws being overshadowed by its strengths. The design of its modules effectively enhances the network's ability to represent multi-scale features.

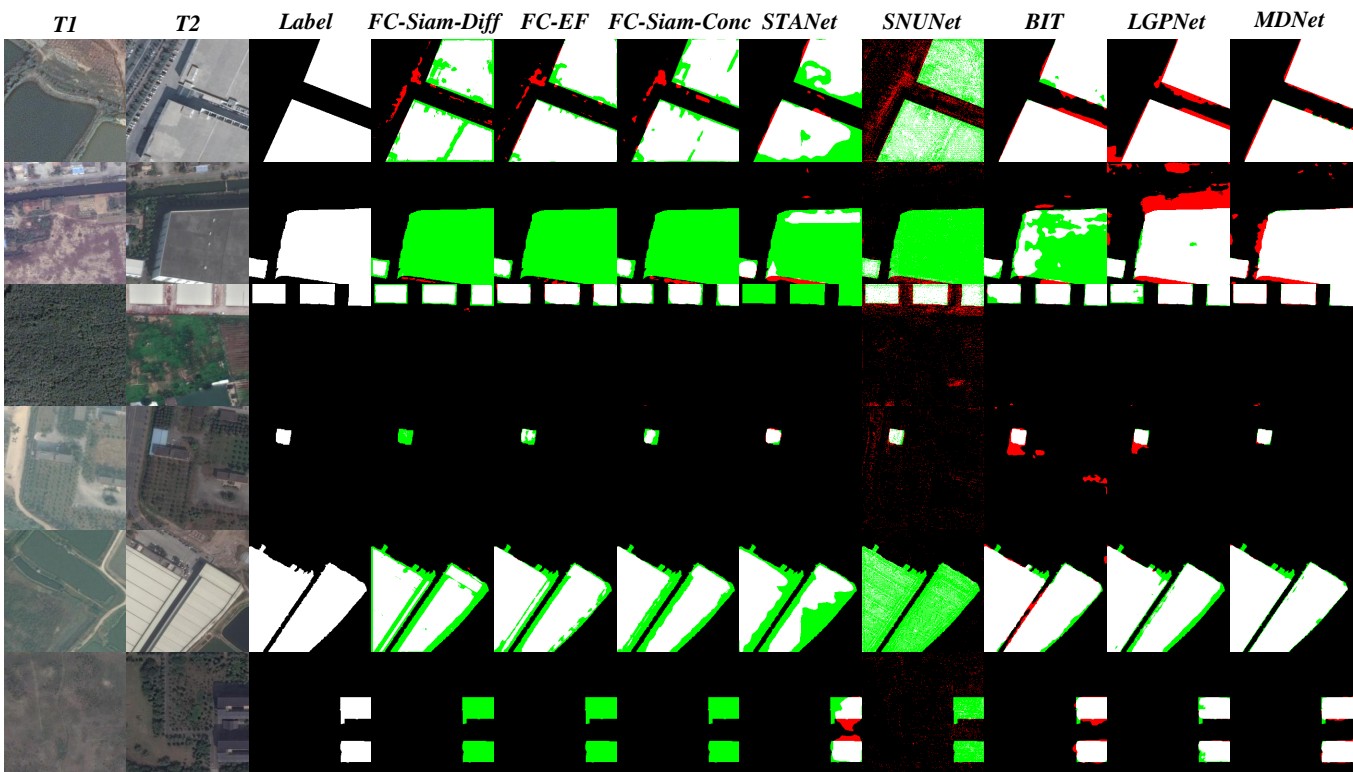

**Figure 6.** Some representative visualization results for the Google Dataset. Each column, from left to right, represents the T1 image, T2 image, ground truth, FC-Siam-Diff, FC-EF, FC-Siam-Conc, STANet, SNUNet, BIT, LGPNet, and our proposed MDNet. Green and red represent missing and false detection pixels, while white and black represent correctly detected changed and unchanged pixels, respectively.

## 5. Discussion

To further investigate the effectiveness of the framework and modules proposed in MDNet, we conducted extensive ablation experiments on MDNet across three benchmark datasets. Tables 4–6 present the ablation experiments of the dual-dimension DCT attention module ($D^3AM$) and multi-scale DCT pyramid (MDP) in MDNet for the WHU-CD, LEVIR-CD, and Google datasets, respectively. The best results are also indicated in bold for all the ablation experiments. Overall, the design of the $D^3AM$ and MDP in the backbone network has led to some improvement in the performance of building change detection. While combining both of them may not have better performance than individual methods for certain challenging datasets, the overall combination of the two proves to be complementary and beneficial. It is worth noting that the improvement in the $D^3AM$ is more significant than that of the MDP. This is because the $D^3AM$ uses DCT to obtain frequency information in both spatial and channel dimensions to refine the feature map, which is considered to be

effective for high-frequency and low-frequency information in building distribution. Both provide better adaptive enhancement. The MDP, on the other hand, further captures multi-scale DCT information to better identify multi-scale land cover targets in VHR images.

**Table 4.** Ablation study of the proposed MDNet on WHU-CD Dataset. The best results are highlighted in bold.

| Methods | Precision | Recall | F1-Score | IOU |
|---|---|---|---|---|
| Backbone | 94.195 | 86.714 | 90.300 | 82.315 |
| Backbone + D$^3$AM | 93.476 | **88.961** | 91.163 | 83.761 |
| Backbone + MDP | 93.528 | 88.493 | 90.941 | 83.387 |
| **Full (MDNet)** | **94.678** | 88.445 | **91.456** | **84.257** |

**Table 5.** Ablation study of the proposed MDNet on LEVIR-CD Dataset. The best results are highlighted in bold.

| Methods | Precision | Recall | F1-Score | IOU |
|---|---|---|---|---|
| Backbone | **91.841** | 88.363 | 90.068 | 81.931 |
| Backbone + D$^3$AM | 91.575 | 89.320 | 90.434 | 82.538 |
| Backbone + MDP | 91.722 | 89.252 | 90.470 | 82.599 |
| **Full (MDNet)** | 91.401 | **89.982** | **90.686** | **82.960** |

**Table 6.** Ablation study of the proposed MDNet on Google Dataset. The best results are highlighted in bold.

| Methods | Precision | Recall | F1-Score | IOU |
|---|---|---|---|---|
| Backbone | 86.906 | 79.292 | 82.925 | 70.830 |
| Backbone + D$^3$AM | 86.339 | **80.895** | 83.528 | 71.715 |
| Backbone + MDP | 87.926 | 79.203 | 83.337 | 71.434 |
| **Full (MDNet)** | **88.356** | 80.359 | **84.168** | **72.664** |

To further demonstrate the effectiveness of our proposed MDP, Figure 7 provides several heatmaps of the spatial attention masks from the proposed D$^3$AM on three benchmark datasets. It can be seen from the figure that some of them can enhance the boundaries of the changed land cover objects; the others can globally depict the whole buildings. This suggests that the D$^3$AM can utilize the DCT to capture global frequency information from low- to high-frequency regions, aiming to enhance feature maps in the network and help improve the recognition capability of land cover targets at multiple scales. Apart from that, MDNet exhibits remarkable extraction capability for the contour features of key changes in the scene with the help of the D$^3$AM. It can be seen from the visualized results that the changed edges of buildings are clearly visible in the extracted features as a whole.

We also provide a comparison of the performance and computational cost of different models, as shown in Table 7. Although the proposed MDNet cannot reach the SOTA in terms of model parameters (Param.) and floating point operations (FLOPs), our model has better performance than models with similar computational costs. Therefore, this compromise solution of sacrificing computational cost to obtain a high-performance model is ideal and has practical significance.

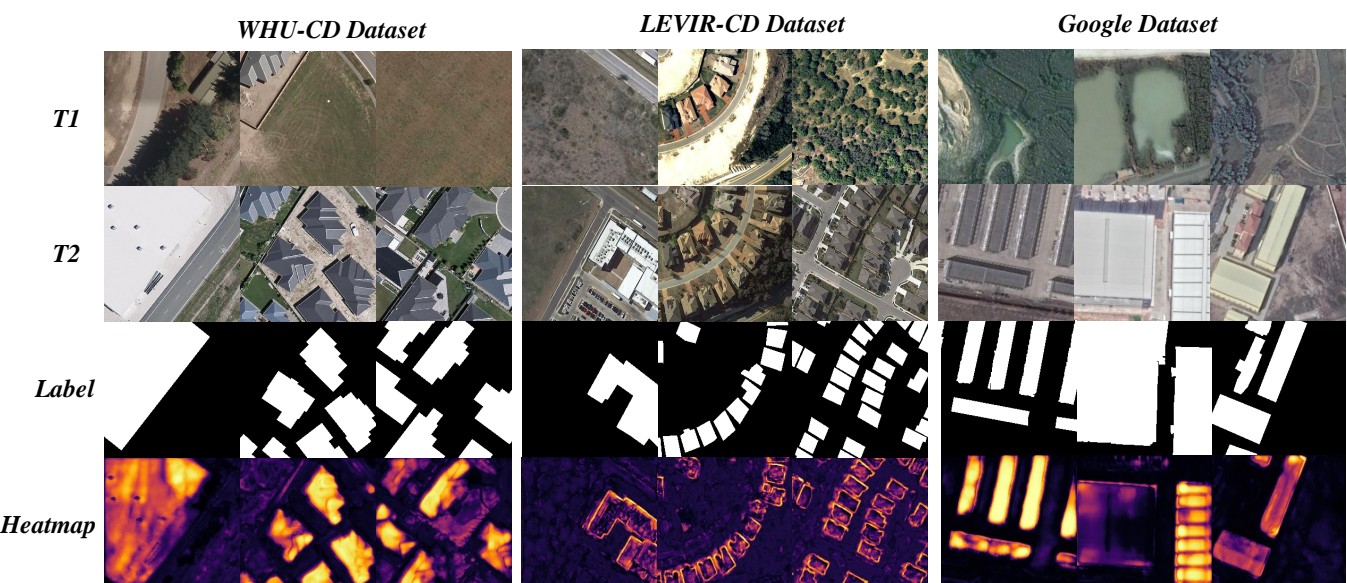

**Figure 7.** The visualization of the heatmaps outputted by the final layer of MDNet on three benchmark building change detection datasets. From top to bottom, each row represents T1, T2, ground truth, and the heatmap of the attention score of the $D^3AM$.

**Table 7.** Quantitative comparison of the performance (in F1-score) and computational costs of different models parameters and FLOPs.

| Methods | Params. (M) | FLOPs (G) | WHU-CD (%) | LEVIR-CD (%) | Google (%) |
|---|---|---|---|---|---|
| FC-Siam-Diff [47] | 21.55 | 18.31 | 76.47 | 84.85 | 54.83 |
| FC-EF [47] | 21.55 | 18.42 | 84.47 | 83.85 | 66.61 |
| FC-Siam-Conc [47] | 24.68 | 18.31 | 64.49 | 84.93 | 65.54 |
| STANet [49] | 16.93 | 6.73 | 85.29 | 84.62 | 63.42 |
| SNUNet [66] | 12.03 | 40.65 | 73.09 | 86.87 | 51.38 |
| BIT [71] | 3.50 | 4.39 | 84.69 | 88.46 | 77.74 |
| LGPNet [57] | 70.99 | 56.66 | 86.58 | 90.01 | 83.19 |
| **MDNet (Ours)** | 55.40 | 59.00 | 91.46 | 90.69 | 84.17 |

## 6. Conclusions

In this paper, a novel multi-scale DCT network (MDNet) is proposed for building change detection in VHR remote sensing imagery. Two crucial components were designed, namely the dual-dimension DCT attention module ($D^3AM$) and multi-scale DCT pyramid (MDP). Specifically, the $D^3AM$ leverages DCT to simultaneously acquire frequency information in both spatial and channel dimensions to refine the feature maps, which is considered to provide better adaptive enhancement for both high-frequency and low-frequency information in building distribution. The MDP further captures multi-scale DCT information to better recognize the multi-scale land cover objects in VHR imagery. Experiments on benchmark datasets indicate that our method performs favorably against state-of-the-art algorithms on numerous benchmark datasets. In addition, the ablation analysis also demonstrates that our $D^3AM$ and MDP modules improve both feature refinement and accurate segmentation of multi-scale targets.

However, the proposed method is built within a supervised learning framework and requires non-negligible time and annotated datasets to acquire the ability to automate building change detection. In this case, the untrained model may have unacceptable performance for change detection tasks. To deal with this problem, unsupervised and self-

supervised learning-based techniques can be helpful for more flexible automated change detection. In future work, we will also actively explore accurate building feature extraction and change information analysis using high-resolution remote sensing data with limited quality, and cross-modal scenarios will also be further considered.

**Author Contributions:** Conceptualization, Y.Z. and L.F.; methodology, Y.Z., L.F. and Q.L.; validation, Y.Z. and L.F.; investigation, Y.Z. and Q.L.; writing—original draft preparation, Y.Z., L.F. and J.C.; writing—review and editing, Y.Z., Q.L. and J.C. All authors have read and agreed to the published version of the manuscript.

**Funding:** This work was financially supported by the Shaanxi Provincial Department of Science and Technology Fund Project "Shaanxi Provincial Innovation Capability Support Program" (No. 2021PT-009).

**Data Availability Statement:** Not applicable.

**Acknowledgments:** The authors thank the anonymous reviewers for their insightful comments and suggestions.

**Conflicts of Interest:** The authors declare no conflict of interest.

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
