# Peer review of "Multi-Scale Discrete Cosine Transform Network for Building Change Detection in Very-High-Resolution Remote Sensing Images"

_remotesensing, doi:10.3390/rs15215243_

Round 1

Reviewer 1 Report

Comments and Suggestions for Authors

A  multi-scale DCT network for building change detection was proposed. Good results achieved. However, the following issues must be addressed:

1.      Title: There are several abbreviations in the title, it is recommended to use the full name.

2.      The line numbers appear to be in the wrong place, and in some places they are discontinuous.

3.      Equation (X) should be mentioned in the main text.

4.      Equation 11 and 12, why are FP’ and FN, not FP and FN?

5.      Moderate editing of English language required.

6.      What is the data set used for this model, and how is it divided?

7.      How stable and robust is your proposed model? How does it look on other data sets? Is as good as on your own dataset. Only from the analysis of the current experimental results, I cannot see the superiority of your model, so it is suggested to increase the evaluation angle of the model.

8.      How is the detection speed? It is recommended to list the results of FPS or inference time.

9.      Limitations of the study should be appropriately mentioned in the conclusion.

Comments on the Quality of English Language

Moderate editing of English language required

Reviewer 2 Report

Comments and Suggestions for Authors

The authors of the submitted paper propose a new deep-learning framework for building change detection. In line with the recent developments, the authors aim to use low-frequency information in the frequency domain for this purpose. The authors claim that their framework, MDNet, offers several novel aspects that result in significant performance enhancement, as seen from implementing the method on three separate datasets.

After reading the paper, this reviewer feels that the way this article is written is not quite the same as what one would expect in a remote sensing journal such as Remote Sensing. Therefore, this work may be placed in a journal more relevant to deep-learning methods. In case the editor and the other reviewers feel that this manuscript is appropriate for consideration Remote Sensing, this reviewer would recommend the following changes and those recommended by the other reviewers and the editor.

  1. Please consider using less jargon (e.g., hand-crafted representation on line 123, or edge-guided recurrent convolutional neural network on like 47); please explain in simple terms what these mean for remote sensing audience and what are their implications for building change detection.
  2. On line 43: What do you mean by the bottleneck in performance?
  3. Please also mention how generalizable your method is, i.e., what if someone were to run this framework for a large area or across cities showing vastly different types of buildings?
  4. Please consider contextualizing your work with more examples and implications using remote sensing examples, essentially bridging the gap between the deep-learning techniques mentioned and their implications for object detection so that the remote sensing audience that may not have a background in deep-learning can appreciate the novelty of your work.

Comments on the Quality of English Language

The quality of the English may be acceptable. However, as mentioned above, please contextualize your work with remote sensing examples and implications so audience not knowledgeable in deep-learning can appreciate and understand the work.

Round 2

Reviewer 1 Report

Comments and Suggestions for Authors

My comments on the initial version of the manuscript have been sufficiently addressed by the authors in this revised version. I have no further comments on the technical aspects. The manuscript may be considered for publication after a proofreading.

Author Response

Thank you again for your professional comments and valuable suggestions. Your comments and suggestions have significantly improved the quality of our paper.

Reviewer 2 Report

Comments and Suggestions for Authors

After reviewing the revised manuscript, this reviewer feels that most of the comments have been addressed by the authors. Hence, subject to other reviewer’s comments and editor’s judgement, this reviewer recommends acceptance of the paper for publication.

However, this reviewer could not find the changes mentioned by the authors in response to this reviewer’s initial comment#5 (as mentioned in the author reply) was implemented on the manuscript. Unless this reviewer missed it, it is brought to the notice of the authors.

Comments on the Quality of English Language

Minor editing may be done.
